# Knowledge, Attitudes and Practices of Veterinarians Towards Antimicrobial Resistance and Stewardship in Nigeria

**DOI:** 10.3390/antibiotics9080453

**Published:** 2020-07-28

**Authors:** Usman O. Adekanye, Abel B. Ekiri, Erika Galipó, Abubakar Bala Muhammad, Ana Mateus, Roberto M. La Ragione, Aliyu Wakawa, Bryony Armson, Erik Mijten, Ruth Alafiatayo, Gabriel Varga, Alasdair J. C. Cook

**Affiliations:** 1Nigeria Ministry of Defence Health Implementation Programme, 900247 Abuja, Nigeria; adekanyeusmanoladipo@gmail.com; 2School of Veterinary Medicine, University of Surrey, Guildford GU2 7AL, UK; e.galipo@surrey.ac.uk (E.G.); R.Laragione@surrey.ac.uk (R.M.L.R.); b.armson@surrey.ac.uk (B.A.); r.alafiatayo@surrey.ac.uk (R.A.); alasdair.j.cook@surrey.ac.uk (A.J.C.C.); 3Life Stock Management Services Limited, 900271 Abuja, Nigeria; bala_bubakar2000@yahoo.com; 4Royal Veterinary College, University of London, London AL9 7TA, UK; amateus@rvc.ac.uk; 5Department of Veterinary Surgery and Medicine, Ahmadu Bello University, 810211 Kaduna, Nigeria; asmwakawa@yahoo.com; 6Zoetis-ALPHA Initiative, Zoetis, B-1930 Zaventem, Belgium; erik.mijten@zoetis.com (E.M.); gabriel.varga@zoetis.com (G.V.)

**Keywords:** antibiotic, antimicrobial resistance, veterinary, animal health, Africa

## Abstract

Antimicrobial resistance (AMR) is a global health concern and the inappropriate use of antibiotics in animals and humans is considered a contributing factor. A cross-sectional survey to assess the knowledge, attitudes and practices of veterinarians regarding AMR and antimicrobial stewardship was conducted in Nigeria. A total of 241 respondents completed an online survey. Only 21% of respondents correctly defined the term antimicrobial stewardship and 59.8% were unaware of the guidelines provided by the Nigeria AMR National Action Plan. Over half (51%) of the respondents indicated that prophylactic antibiotic use was appropriate when farm biosecurity was poor. Only 20% of the respondents conducted antimicrobial susceptibility testing (AST) frequently, and the unavailability of veterinary laboratory services (82%) and the owner’s inability to pay (72%) were reported as key barriers to conducting AST. The study findings suggest strategies focusing on the following areas may be useful in improving appropriate antibiotic use and antimicrobial stewardship among veterinarians in Nigeria: increased awareness of responsible antimicrobial use among practicing and newly graduated veterinarians, increased dissemination of regularly updated antibiotic use guidelines, increased understanding of the role of good biosecurity and vaccination practices in disease prevention, and increased provision of laboratory services and AST at affordable costs.

## 1. Introduction

Resistance to antimicrobials is rising worldwide, threatening our ability to treat common infectious diseases of humans and animals [1]. The direct consequences of infection with resistant microorganisms can be severe, including longer and more severe illnesses, increased mortality, prolonged stays in hospital, increased rates of therapeutic failure resulting in loss of protection for patients undergoing operations and other medical procedures, and increased healthcare costs [2]. The overuse and misuse of antibiotics in humans and animals has been linked to the emergence of antimicrobial resistance (AMR) in animals and the environment [1,3,4,5,6,7]. In livestock farming, antibiotics are used for prophylaxis, metaphylaxis (the treatment of a group of animals after the diagnosis of infection and/or clinical disease in part of the group), growth promotion, or are used therapeutically, to maintain health and increase productivity. Interaction between animals, humans and the environment promotes the transfer of resistant genes across different species, making AMR an important One Health challenge emerging on a global scale [3,4,5,6,8]. With the dwindling repertoire of antibiotic options available for the control of emerging, life-threatening and multi-drug resistant bacteria, there is a need for proper antibiotic stewardship to preserve the efficacy of existing antibiotics [9]. 

Antibiotic stewardship programs can play an important role in improving, prescribing and optimizing the use of antibiotics [10,11,12,13,14]. In the human sector, which has made significant strides in this area, antibiotic stewardship programs are defined as hospital-based programs dedicated to improving antibiotic use [15]. These programs increase infection cure rates while reducing treatment failures, adverse effects, hospital costs and lengths of stay, and antibiotic resistance [14,16,17]. Considering the potential benefits of antibiotic stewardship programs, the World Health Organization (WHO) strongly recommends that governments implement them for the containment of AMR [18]. Therefore, it is imperative that governments implement tailored interventions to encourage antimicrobial stewardship among healthcare professionals [19]. Beyond stewardship programs, strategies to tackle other related challenges also need to be considered. 

In most sub-Saharan African countries, surveillance programs for antibiotic use and AMR in humans and animals are either lacking or are in their infancy, and the human and animal healthcare sectors at the government or ministry levels tend to work in silos, resulting in a lack of intersectoral collaboration. Furthermore, in Nigeria, the lack of regulation of existing veterinary drug markets and low involvement of pharmacists [20] and veterinarians in the formal drug distribution market [21] may contribute to the issue of substandard drugs in the marketplace. In 2018, the Director General of the Nigeria National Agency for Food and Drug Administration and Control (NAFDAC) held a town hall meeting with all players in the livestock industry, including practicing veterinarians and manufacturers and suppliers of veterinary medicines, and at the meeting announced the ban of use of some antimicrobials and growth promoters in livestock as part of efforts to control AMR and to support the One Health triad [22]. The banned antimicrobials and growth promoters included chloramphenicol, furazolidone, metronidazole, nitrofuran, and carbadox [22]. To address AMR in both humans and animals in sub-Saharan African countries, strong multidisciplinary collaborations are needed; however, these are lacking because of the poor One Health coordination of animal and human national disease surveillance systems [23].

In 2017, the Nigerian Federal Ministry of Agriculture and Rural Development (FMARD), Federal Ministry of Environment and Federal Ministry of Health developed a National Action Plan (NAP) for antimicrobial resistance (the Nigeria Center for Disease Control’s five-point action plan) as part of the country’s efforts to address the problem of AMR and to promote the responsible use of antimicrobials through a One Health approach [23]. Veterinary surgeons are typically responsible for prescribing and overseeing antimicrobial use in animals. Therefore, the role of the veterinarian in tackling AMR cannot be over-emphasized as they are the custodians of antimicrobials used in animal health [24] and food production. In Nigeria, regulatory authorities like NAFDAC and the Nigeria Centre for Disease Control (NCDC) are involved in creating awareness among veterinarians and veterinary students on the challenge of AMR through campaigns convened by the Veterinary Council of Nigeria (VCN) and the umbrella association for veterinarians in Nigeria, the Nigeria Veterinary Medical Association (NVMA). 

Despite the potential negative impact of AMR on animal and public health, there remains a paucity of data concerning the awareness of this problem in sub-Saharan countries [25]. The attitudes of veterinarians towards antibiotic use and determinants influencing prescribing behavior of veterinarians have been investigated elsewhere [26,27,28,29]. In Nigeria, a few studies have explored the knowledge, attitude and practices of veterinarians towards antibiotic use, resistance and stewardship. A previous study by Anyanwu and Kolade reported that knowledge about antibiotic stewardship among veterinarians was as low as 21.4% [30]. However, this study was limited to Enugu State and used a non-probability sampling technique, which affected the generalizability of the study findings [30]. A recent study that involved only veterinary students reported that 60% of respondents had unsatisfactory knowledge scores for AMR [31]. To expand on the knowledge base in this area and to inform the development of interventions to promote responsible antibiotic use, a nationwide study of veterinarians in Nigeria was conducted. The objective of this study was to assess the knowledge, attitudes and practices towards AMR and antimicrobial stewardship and to identify factors that influence antibiotic prescription practices of veterinarians in Nigeria.

## 2. Results

The survey was sent to 5603 participants; there were 488 responses, corresponding to a response rate of 13%. Out of the 488, six did not consent, 59 consented but did not start or attempt the survey, while 128 consented and attempted but did not complete the survey. Thus, 241 respondents consented and completed the survey. 

### 2.1. Demographic Information

Most of the respondents were male (79.7%, 192/241) and almost half were aged 25-34 years (48.1%, 116/241) (Table 1). A majority of respondents reported having a veterinary degree (63.1%, 152/241) as the highest educational qualification. More than a third of respondents had been registered as a veterinarian for 0–5 years (36.1%, 87/241). Mixed practice (defined as any combination of small, large, poultry or other type of practice) was the most frequently reported type of practice (63.5%, 153/241). Five percent of respondents (5.4%, 13/241) did not practice. Most of the respondents were employed in private practice (38.2%, 92/241).

The distribution of respondents based on the location of the vet practice within Nigeria’s six geopolitical zones showed that almost half of the respondents were located in the North Central (29%, 69/241) and the North West zones (20%, 49/241) (Table 1). The lowest number of respondents was recorded in the South South zone (7%, 17/241). The distribution of respondents based on the location of the vet practice within Nigeria’s 37 states indicated that the Federal Capital Territory, Abuja, had the highest number of respondents (13.7%, 33/241) followed by Kaduna State (10%, 24/241) and Borno State (7.1%, 17/241) (Figure 1). Yobe and Nasarawa States were the only two states without any respondents.

### 2.2. Knowledge

Eighty-nine (36%) of the 241 respondents had heard of the term antimicrobial stewardship and of these, 69% (61/89) were able to correctly define antimicrobial stewardship (Appendix A). Most of the respondents (81.7%, 197/241) were able to differentiate between an antibiotic and an antimicrobial agent. Most of the respondents were aware that antibiotics kill both commensal and pathogenic bacteria (91.3%, 220/241) and 94.6% (228/241) knew that overuse of antibiotics renders them ineffective. Most respondents were aware that antibiotics do not kill viruses (93.4%, 225/241) and all respondents were aware that bacteria can become resistant to antibiotics. Many respondents (93.4%, 225/241) were aware that there was a need to observe withdrawal periods before consuming milk from cows treated with antibiotics, and 97.9% (236/241) were aware that a withdrawal period is necessary before treated poultry can be considered fit for human consumption. 

More than half of the respondents (59.8%, 144/241) had not heard of or read the Nigeria Center for Disease Control’s five-point action plan for responsible use of antimicrobials. When asked what topics they would like to receive more information on, 75.5% (182/241) of the respondents selected “links between the health of humans, animals and the environment”, and 55.6% (134/241) chose “microbial culture and sensitivity testing” (Appendix A). 

### 2.3. Attitude

All but one (99.6%, 240/241) of the respondents believed that veterinarians have a role to play in preventing public health threats posed by AMR (Appendix A). Respondents were asked to indicate if they considered the following as important global challenges and the most frequently reported challenges were: AMR (80.9%, 195/241), food security (74.5%, 180/241) and climate change (60.9%, 146/241). Most respondents (97.9%, 233/238) thought AMR was a national problem in Nigeria, and 96.1% (223/232) of respondents believed AMR will be a greater problem in veterinary practice in the future than it is today. Note the denominators used to calculate the above two percentages, and in the sections below (where relevant), do not add up to 241 because they exclude “unknown” responses (Appendix A).

Many of the respondents considered the excessive use of antibiotics in livestock (83.8%, 202/241) and under dosing of antibiotics (78.8%, 185/241) as the most important potential contributors to the development of AMR (Appendix A). Most of the respondents agreed that prescribing unnecessary antibiotics was professionally unethical (97.5%, 234/240) and 78.2% (172/220) believed the antibiotics they prescribe may contribute to AMR (Appendix A). Most respondents (99.2%, 237/239) agreed with the statement “biosecurity was important in food production” and 28.2% (68/241) considered poor biosecurity practices as a contributor to AMR development. Of concern was that over half of the respondents (56%, 112/200) agreed with the statement “Prophylactic antibiotics are an appropriate alternative to protect animal health when there is poor biosecurity”. Almost half of the respondents (42.3%, 102/219) indicated that they lacked enough knowledge on antibiotic use, while 28.9% (59/204) believed there were not enough antibiotics under development to combat the problem of resistance (Appendix A). 

### 2.4. Practices Influencing Antibiotic Use

A total of 132/220 (60%) of the respondents reported that they frequently encountered animal owners who had already initiated antibiotic treatment without veterinary supervision (Appendix A). Note the denominators used to calculate the percentages for some variables in this section (where relevant) do not add up to 241 because they exclude “don’t know” responses (Appendix A).

More than half of the respondents (68.4%, 154/225) reported that their practice had a standardized protocol for the treatment of sick animals (Appendix A). When asked what guidelines were followed to help select the appropriate antibiotic when a patient was presented for the first time, 53.1% (128/241) of respondents reported using microbiological culture and antimicrobial susceptibility testing (AST) for guidance. More than a half of the respondents (59.8%, 144/241) indicated that they administered empirical treatment while awaiting AST results. A small proportion (14.5%, 35/241) reported selecting antibiotics based on what the client could afford. 

With regards to the frequency of AST use before starting antibiotic treatment, 48.7% (112/230) of respondents used AST at a frequency of one to three times in a month while only 21.3% (49/230) used AST more than three times a month. Over a quarter of the respondents (30%, 69/230) never conducted AST (Appendix A).

Most respondents (75.5%, 182/241) indicated that poor response to initial antibiotic treatment or treatment failure influenced the veterinarians’ decision to request AST. Other reported drivers for AST use included recurrent health conditions (70.5%, 170/241), having no knowledge of the animal or farm’s health history (26.6%, 64/241) and owner request (18.3%, 44/241) (Appendix A). When asked what were the most important barriers to the use of AST, the majority of respondents selected unavailability of laboratory services (82.2%, 198/241), followed by owners’ inability to pay for AST tests (71.8%, 173/241), urgent need for antibiotic therapy (56.8%, 137/241), long waiting time for AST results (35.3%, 85/241) and uncertainty of what to request from the lab to guide antibiotic selection (3.3%, 8/241) (Appendix A). 

The cost of antibiotics (80.9%, 195/241) and owners’ ability to pay (81.3%, 196/241) were reported to influence the respondent’s decision when selecting antibiotics. Other cost related influences included: expected profit margin to the veterinarian (27.4%, 66/241), marketing offers (16.6%, 40/241), adverts by pharmaceutical company representatives (12.9%, 31/241) and medicine sellers (8.3%, 20/241) (Appendix A). 

When asked what antibiotic characteristics had the most influence upon the veterinarian’s selection of antibiotics, 85.5% (206/241) of respondents reported the antibiotic’s spectrum of activity, 63.1% (152/241) reported AST results, 50.6% (122/241) reported withdrawal period, 44.4% (107/241) reported the ease of administration and 43.2% (104/241) reported the risk of development of AMR (Appendix A). Other reported factors that influenced veterinarians’ decision to select antibiotics were veterinarians’ previous experience (96.3%, 232/241), advice from colleagues (68%, 164/241) and owner preference for a specific antibiotic (9.5%, 23/241) (Appendix A). 

Finally, when asked what sources of information influenced the veterinarians’ decision the most when selecting an antibiotic to use, 79.3% (191/241) of respondents indicated veterinary education and training, followed by prescription guidelines or policies supplied by veterinary hospital or bodies (68.9%, 166/241), product labels or leaflets (64.3%, 155/241), legal restriction of drug to a defined species (38.6%, 93/241) and published scientific literature (35.3%, 85/241) (Appendix A).

### 2.5. Relationship between the Use of AST before Antibiotic Treatment and Select Investigated Parameters

The relationship between the “use of AST before antibiotic treatment” and nine selected variables was assessed (Appendix A). The analyses included 230 respondents that responded to the variable “use of AST before antibiotic treatment” and excluded respondents that answered “don’t know” (n = 11). The proportion of respondents that reported the “use of AST before antibiotic treatment” was significantly different across the response levels for the following variables: years in practice (P = 0.049), knowledge of correct definition of antimicrobial stewardship (P = 0.032), knowledge of NCDC five points (P = 0.003), agreement with the statement that prophylactic use of antibiotics when farm biosecurity is poor is inappropriate (P = 0.029), and having a standard antibiotic treatment protocol in the veterinary practice (P ≤ 0.001) (Appendix A). 

### 2.6. Relationship between Knowledge Level of Appropriate Antibiotic Use and AMR and Select Investigated Parameters

The relationship between “knowledge level on appropriate antibiotic use and AMR” and selected variables was assessed (Appendix A). The analyses included 240 respondents that responded to the variable “knowledge level on appropriate antibiotic use and AMR” (assigned to the category “high knowledge” or “low to moderate knowledge”) and excluded one respondent that was assigned a knowledge score of zero because they had provided no correct answer. The proportion of respondents with high knowledge was significantly different across the response levels for the following variables: age group (P = 0.024), education level (P = 0.024), and agreement with the statement that prophylactic use of antibiotics when farm biosecurity is poor is inappropriate (P = < 0.001) (Appendix A). 

## 3. Discussion

The current study assessed the knowledge, attitudes, and practices towards AMR and antimicrobial stewardship of veterinary professionals in Nigeria. To the best of our knowledge, this is the first nationwide baseline study on the subject. Most of the veterinarians in the current study were between 25 and 44 years old, which probably explains why over 67% of respondents were within 10 years of being registered veterinary surgeons. Most of the respondents were private or government practitioners (69%), followed by teaching (10.4%) and non-governmental organization employees (9.1%); these results likely reflect the distribution of veterinarian employment in Nigeria.

Sixty three percent of the respondents were familiar with the term antimicrobial stewardship, compared to 17% reported in a similar study conducted in Enugu State, Nigeria [30]. The reasons for the observed differences are not clear but the current study can be considered more representative because it targeted participants across the country. However, our study highlights that there is still inadequate awareness of the concept of antimicrobial stewardship among veterinarians. Our study findings also revealed that the proportion of respondents who used AST before antibiotics administration varied among respondents who correctly or incorrectly defined antimicrobial stewardship. This finding suggests that educational strategies aimed at increasing awareness of antimicrobial stewardship among practicing and new veterinarians both at the practice and veterinary school levels may be helpful in promoting the responsible use of antibiotics.

Although most respondents reported that antibiotic resistance occurred in bacteria (98%) and could differentiate between an antibiotic and an antimicrobial (82%), a small percentage of respondents (4%) reported that antibiotics kill viruses, suggesting this proportion of respondents may prescribe antibiotics for viral infections. In comparison, 1% of student healthcare professionals (human and animal health students) surveyed in the United Kingdom thought antibiotics killed viruses [32], suggesting that study population may have been more knowledgeable on this aspect compared to our study population. Additionally, in our study, a small percentage of respondents (6%) reported that a withdrawal period of antibiotics-treated animals is not necessary before milk consumption. Failure to observe appropriate withdrawal periods following antibiotic treatment may result in the introduction of antibiotic residues in animal foods consumed by humans [33,34]. The failure to observe appropriate withdrawal periods has potential human health implications including the development of drug-related allergies and hypersensitivity reactions, especially with beta lactam antibiotics and penicillin [35], and the risk of development of AMR [36,37]. 

Although most respondents knew that biosecurity is important in food production, this study highlighted a misconception regarding the link between biosecurity and antibiotic use. Over half of the respondents thought that prophylactic antibiotic use was appropriate in situations where biosecurity was poor. The reliance on antibiotics when biosecurity practices are poor has been reported in other studies [38]. Antibiotics can be an integral part of disease preventive methods but should be used only when indicated; they should not be used as the first line of action [39] or as a substitute for poor biosecurity practices [40]. Prophylactic and metaphylactic use of antibiotics administered to animal groups through water and feed may lead to increased environmental concentration of antibiotic residues which can in turn result in exposure of animals and humans alike and elevate the risk of AMR development [41,42]. Based on the current study findings there is a need for an improved understanding among veterinarians of the role of biosecurity practices in preventing and minimizing the risk of infections and reducing the overuse of antibiotics. Biosecurity practices are a key component of animal husbandry and disease prevention measures that can be implemented to improve animal health and welfare, and to reduce the need to use antimicrobials [43]. 

This study showed that the proportion of respondents with a high knowledge score varied with age group and education level. Although not conclusively established in the current study, this suggests that it is possible that older respondents and those with additional training to a veterinary degree may have a higher knowledge of appropriate antibiotic use and AMR or that the observed differences may be linked to work experiences accrued over time. Further investigation and understanding of the perceptions and barriers to responsible use of antibiotics among this subpopulation would help inform educational efforts. 

More than half of the respondents’ practices had a standard protocol for the treatment of sick animals to ensure the correct dosages and regimes are administered and to reduce the risk of adverse drug reactions or drug toxicity in the animals. Additionally, the proportion of respondents that reported the use of AST before antibiotic treatment was higher among those with a standard treatment protocol compared to those without. The observed relationship suggests having a standard treatment protocol may contribute to good antimicrobial stewardship. In the human sector, the use of standard treatment guidelines is considered an effective means of improving patient care while enhancing cost savings and changing behavior [44]. The treatment guidelines also reflect data on antimicrobial resistance, recognizing that local patterns of resistance often differ across geographical regions [44]. In the context of the animal health sector in Nigeria, strategies that consider up-to-date antimicrobial stewardship guidelines may be helpful in promoting appropriate antibiotic use among veterinary professionals. 

Over half of the respondents (53%) reported conducting microbiological culture and AST before starting treatment, which is higher than the 24% reported in a study conducted in Enugu State, Nigeria [30], and the 38% reported among veterinarians in Europe [45]. The reason for these differences is not clear but may be related to regional differences in the level of awareness of the need to conduct microbiological culture and AST before starting antibiotic treatment or reporting bias which can occur in questionnaire-based studies. However, when asked specifically how often AST was requested before starting antibiotic treatment, only 20% of respondents reported requesting AST frequently (more than three times a month) and a third of the respondents never conducted or requested AST before commencing treatment. The observed drop from 53% to 20% may also be explained by reporting bias. Nevertheless, these findings suggest a considerable number of veterinary professionals do not use AST. There is a need to examine the reasons for the low AST use and identify appropriate interventions.

The respondents in the current study reported that poor response to initial antibiotic treatment (76%) and conditions that recur (71%) were the main factors that influenced their decision to conduct AST, but a lack of information on the animal or farm’s health status, and owner requests were also influences, consistent with the findings from a study conducted in Europe [45]. The findings suggest having structured and up-to-date antibiotic use guidelines at the practice level, having access to rapid, cheap diagnostic tools and being able to handle clients’ expectations through effective communication, may be helpful in providing guidance to veterinary professionals.

Several barriers to the use of AST were reported in this study. The unavailability of laboratory services and owner’s inability to pay were reported as key barriers to the use of AST. The owner’s inability to pay is a major concern because it limits the options available for a veterinarian in making decisions that allow for an appropriate diagnosis and treatment and may subsequently negatively influence the veterinarian’s decisions and choices regarding antibiotic use. Strategies that explore ways to increase availability of veterinary laboratory services across the country and the provision of AST at affordable costs are necessary. Additionally, respondents reported that there is often an urgent need to administer antibiotics due to the acute onset of severe clinical signs. This is an accepted practice when antibiotics are urgently needed to counteract the disease progression and when delays in administering the therapy can lead to a poor outcome [46]. Nonetheless, veterinarians should take several factors into account to inform their decision on a sound empirical therapy: records from previous AST results in the local area if available, information on local patterns of bacterial resistance if available, previous patient cultural and AST results, the suspected anatomic site affected by infection and etiologic pathogen [47]. The records from previous AST results in the local area and information on local patterns of bacterial resistance were unlikely to be available in the context of Nigeria, emphasizing the need for a national integrated AMR surveillance program, as identified by the Nigeria National Action Plan for Antimicrobial Resistance [23]. The empirical antibiotic treatment should not prevent veterinarians from submitting samples for AST but instead be considered a temporary intervention while waiting for AST results that will inform the final, targeted, antibiotic treatment [46].

Veterinary education or training followed by prescription guidelines and policies were the most frequently selected parameters that influence a veterinarian’s decision to select antibiotics. These findings suggest the veterinary curriculum may be a useful means to provide training on appropriate antibiotic use and selection. For example, in the fourth year of study, veterinary students in Nigeria undertake a course in pharmacology and therapeutics, which involves instruction on the types of veterinary pharmacological products and prescription practices. Furthermore, a relationship was observed between having additional education or training on AMR and knowledge of appropriate antibiotic use. Veterinary education or training may present an opportunity to expand and strengthen knowledge on appropriate antibiotic use practices and antimicrobial stewardship, if included in the training curriculum for new animal health professionals. Veterinary education or training also provides an opportunity for practicing veterinarians to update their knowledge, as prescription practices and protocols change over time.

The finding of antibiotic prescription guidelines as one of the most frequently selected parameters that influence a veterinarian’s decision to select antibiotics suggests that there is a need for updated and increased dissemination and uptake of antibiotic guidelines. The Nigerian Veterinary formulary [48], provided by the Veterinary Council of Nigeria (VCN) to guide the prescription and administration of veterinary pharmaceuticals in different animal species was produced in 2007 and has not been updated since. In the present study, less than half of the respondents were aware of the Nigeria Center for Disease Control (NCDC) five-point agenda for AMR control, and a relationship was observed between awareness of the NCDC five-point action plan and AST use. These findings suggest regular updating of the VCN guidelines combined with increased awareness of the NCDC five-point action plan may be helpful in promoting antimicrobial stewardship among veterinary professionals. 

A concerning, but not surprising, observation was that 86% of respondents reported encountering client-initiated antibiotic therapy without veterinary supervision. Even though the current regulatory policies in Nigeria require that only qualified veterinarians and para-vets can administer medications and treatment, livestock farmers can obtain and administer antibiotics without the requirement of a veterinarian’s prescription and this is most likely to occur without regard to antibiotic indication guidelines [49]. This highlights the need for education of not just the veterinary professionals but also the clients, veterinary drug sellers or shop keepers, pharmacies and farmers on appropriate antibiotic use and the risk of antibiotic misuse and AMR. In addition, government interventions such as the formulation and implementation of relevant policies and regulations may also be useful in improving appropriate antibiotic use and stewardship.

In the present study, the proportion of respondents that used AST before antibiotic treatment varied with knowledge of antimicrobial stewardship, knowledge of NCDC’s five points, and with agreement with the statement that prophylactic use of antibiotics is inappropriate when biosecurity is poor. These findings suggest that knowledge of antimicrobial stewardship, NCDC’s five points and prophylactic use of antibiotics may be related to appropriate antibiotic use. These areas could be targeted when developing strategies to improve antimicrobial stewardship and reduce AMR in veterinary practice in Nigeria.

A few limitations were observed during the conduct of the current study. The response rate of the survey was low, and as such, the study’s findings may not be generalizable to the whole of the country. Nevertheless, the gaps identified can still be used to inform discussions by policy makers involved in the development of interventions targeting all veterinarians in Nigeria. The low response rate may have been due to several factors such as unwillingness to participate or lack of internet access in some parts of the country. There may also have been selection bias in the respondent population. For example, it is possible that respondents that completed the survey were more technologically astute or inclined. It could also be that these respondents had a special interest in the subject, hence their participation. Another potential limitation was social desirability bias which might have affected the nature of the responses provided; it is possible that some respondents may have declined to share information they considered inappropriate or erroneous, resulting in an under-reporting of certain aspects on antibiotics and AMR knowledge and practices. Finally, it is important to note that the data analyses performed to assess relationships between selected variables in the current study were exploratory in nature and were not intended to be exhaustive, therefore, no additional inferential analyses such as logistics regression were conducted.

## 4. Materials and Methods 

Ethical review and approval were granted by the Nigeria Ministry of Defense Health Research Ethics Committee, Abuja, via an ethics review application (Ethics approval number: MODHREC/APP/20/12/11/20/1/8/) and by the Research Integrity and Governance Office at the University of Surrey, United Kingdom (Response ID: 353003-352994-41119696).

### 4.1. Study Area

Nigeria is the most populous nation located in West Africa, with an estimated population of approximately 202 million people [50]. Crude oil, agriculture and solid minerals are the mainstay of the economy. It has 37 states including the Federal Capital Territory, Abuja. There are three major tribes and about 250 ethnic groups. Nigeria has a tropical climate and two distinct weather seasons (rainy and dry seasons). To the north of the country is the Sahel climate, to the west is the tropical savannah while the south and east are characterized by tropical monsoon climates.

### 4.2. Study Population

There are circa 8000 registered veterinarians in Nigeria involved in livestock/large animal practice, small animal/companion animal practice, poultry practice, public health and academia (personal communication with Interim College secretary, College of Veterinary Surgeons, Nigeria). This study involved veterinarians registered with the Veterinary Council of Nigeria (VCN). For this study, a registered veterinarian was an individual who obtained certification from the VCN as a Doctor of Veterinary Medicine (DVM) upon completion of the six-year university degree program in Nigeria. All registered veterinarians were pooled together irrespective of the type of practice. 

### 4.3. Study Design

This was a cross-sectional study; a questionnaire survey was designed and used to collect data on the knowledge and attitudes towards AMR and antimicrobial stewardship of veterinarians during the period January to February 2019.

### 4.4. Sample Size

Sample size was estimated as described by Lwanga and Lemeshow (1990) [51] with the prevalence of knowledge of antibiotic resistance set at 50% based on a previous study that investigated the veterinary drugs market in Nigeria [21] and the desired level of precision set at 0.02. The estimated calculated sample size was 2400. A non-response rate of 30% was estimated because studies have shown response rates to web-based surveys are generally low [52] and assumed some respondents would have poor internet access or did not consent. The final estimated minimum sample size considering the non-response rate was 3120 respondents (2400 × 1.3).

### 4.5. Data Collection

A questionnaire was developed and select questions from previous studies [32,45] were adapted to collect data on demographics, knowledge, attitudes, practices towards antibiotic use and resistance, and awareness of antimicrobial stewardship (Appendix A). The tool was pretested among ten veterinarians in Abuja and thereafter questions were further refined to produce the final survey. The final questionnaire was administered electronically using the Qualtrics^®^ survey platform. 

The available list of 5800 registered veterinarians in Nigeria was obtained from the VCN, which was more than the calculated minimum sample size of 3120 respondents. A total of 5603 with phone numbers were randomly selected from this pool of 5800 using a table of random numbers. A link to the survey was sent via a text message to all the 5603 contacts. Of the 5603 contacts, 2662 had email addresses on the VCN register, and the survey link was sent to them via email (in addition to the text message sent to all 5603 contacts). The message inviting contacts to participate in the survey was endorsed by the Nigeria Center for Disease Control (NCDC) and Federal Ministry of Agriculture and Rural Development (FMARD). Further emails and SMS reminders were sent 2, 4 and 6 weeks after the initial message. The survey was made available online for 8 weeks between 2nd January and 28th February 2019. 

### 4.6. Data Analysis

Survey results were downloaded from Qualtrics^®^ to Microsoft Excel. Data collected during the piloting of the survey were excluded from the final analysis. Descriptive statistics were used to summarize the data using R-Studio 1.2.1335.0. 

As part of the descriptive analysis, a scoring system was used to assess the knowledge level on antibiotic use and AMR. A set of nine survey questions on knowledge was selected, and for each question, a score of one point was assigned for each correct answer, with a maximum of nine points allowed (Appendix A). One respondent that provided no correct answer was assigned a score of zero. Respondents were further regrouped into two categories based on knowledge score; respondents scoring ≥ 7/9 points were assigned to the category “high knowledge” and those scoring < 7/9 points were assigned to the category “low to moderate knowledge”.

Following the descriptive analysis, a bivariate analysis was performed using the Chi-square test or Fisher’s exact test, as appropriate, to explore the relationship between two selected outcome variables and eleven selected variables. The outcome variables were: “the use of AST before antibiotic treatment” and “knowledge level on appropriate antibiotic use and AMR”. The relationships between these two outcome variables and the following 9 variables were assessed: age group of participants, gender, educational level, years in practice, type of practice, practice location, knowledge of the NCDC five-point action plan for correct antibiotic use, prophylactic use of antibiotic when biosecurity is poor, and existing antibiotic treatment protocol in practice. Two further variables, the correct definition of antimicrobial stewardship and owner-initiated treatment, were investigated for the outcome variable “the use of AST before antibiotic treatment”. Two additional variables were also investigated for the outcome variable “knowledge level on appropriate antibiotic use and AMR”: type of employment and knowledge of antimicrobial stewardship. A two-tailed P-value of ≤0.05 was considered statistically significant. It is important to note that the association analyses were exploratory in nature, used selected variables and were not intended to be exhaustive, therefore, no advanced additional inferential analyses, such as logistics regression, were conducted.

## 5. Conclusions

Findings from this study provided baseline evidence on the knowledge, attitudes and practices regarding antibiotic use, AMR and antimicrobial stewardship among veterinarians across Nigeria. With respect to knowledge and attitudes on appropriate antibiotic use and AMR, there was little awareness of the concept of antimicrobial stewardship among veterinarians, and the role and use of biosecurity, as well as the prophylactic antibiotic use in the prevention of infection, were not well understood. There is a need for an increased understanding among veterinarians for how the use of biosecurity practices plays a role in the prevention of infection, reducing the burden of disease in animal populations and, therefore, in reducing the need for and use of antibiotics. Education or training strategies aimed at increasing awareness of antimicrobial stewardship among practicing and new veterinarians at the practice and veterinary school levels may be helpful in promoting antimicrobial stewardship. 

Regarding practices and factors influencing antibiotic use, the use of AST to inform antibiotic treatment was low, suggesting a need to further examine the reasons for this and identify appropriate interventions. The unavailability of laboratory services and the client’s inability to pay were reported as key barriers to AST use. Strategies that explore ways to increase the availability of veterinary laboratory services across the country and the provision of AST at affordable costs are necessary. 

Veterinary education or training followed by prescription guidelines and policies were the most frequently selected parameters that influence a veterinarian’s decision to select antibiotics. The regular updating of the antibiotic prescription and use guidelines combined with increased awareness and dissemination among veterinarians may be helpful in promoting antimicrobial stewardship. Finally, the reported client-initiated antibiotic therapy was also a concern highlighting the need for education of not just the veterinary professionals, but also their clients and drug shop keepers on appropriate antibiotic use and stewardship.

## Figures and Tables

**Figure 1 antibiotics-09-00453-f001:**
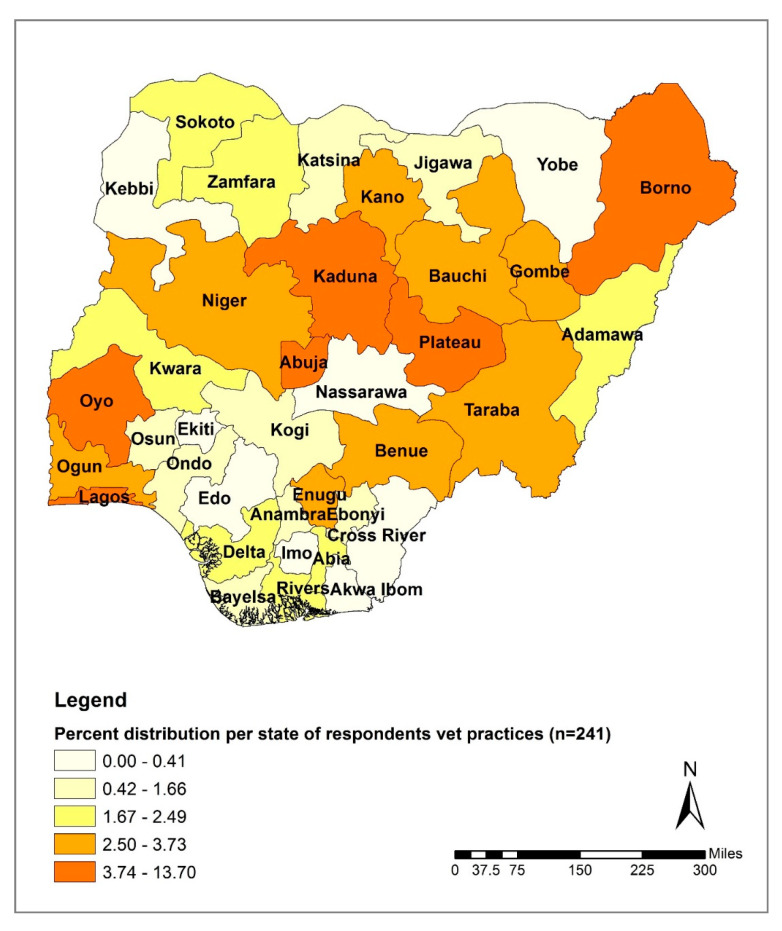
Distribution of survey respondents based on reported location of veterinary practice by state in Nigeria.

**Table 1 antibiotics-09-00453-t001:** Demographics of respondents

Variable	Response	Frequency (n = 241)	Percentage (%)
Gender	Female	49	20.3
	Male	192	79.7
Age group	18–24 years old	1	0.4
	25–34 years old	116	48.1
	35–44 years old	85	35.3
	45–54 years old	27	11.2
	55–64 years old	12	5.0
	65 years and above	0	0
	Prefer not to say	0	0
University of training	Ahmadu Bello University (ABU), Zaria	69	28.6
	University of Maiduguiri	60	24.9
	University of Nigeria Nsukka (UNN)	31	12.9
	Usmanu Danfodiyo University, Sokoto (UDUS)	27	11.2
	University of Ibadan (UI), Ibadan	27	11.2
	Federal University of Agriculture Makurdi (FUAM)	9	3.7
	Federal University of Agriculture, Abeokuta (FUNAAB)	8	3.3
	Other	8	3.3
	University of Abuja	2	0.8
Highest level of education	Veterinary degree (DVM, etc.)	152	63.1
	MSc/MPH	61	25.3
	Fellow, College of Veterinary Surgeon (FCVS)	6	2.5
	PhD	16	6.6
	Other	6	2.5
Type of employment	Private practice	92	38.2
	Government employee	75	31.1
	Teaching	25	10.4
	Non-governmental organization employee	22	9.1
	Research	15	6.2
	Other	12	5.0
Years registered as a vet surgeon	0–<5 years	87	36.1
	5–10 years	75	31.1
	10–15 years	30	12.4
	15–20 years	18	7.5
	21 and above	28	11.6
	Prefer not to say	3	1.2
Type of veterinary practice *	Mixed practice (large, small or exotic animals)	97	40.2
	Poultry practice (chicken, turkey)	94	39.0
	Small animal practice (dogs, cats, rabbits)	87	36.1
	Large animal practice (cattle, horse, goat, sheep, pig)	75	31.1
	Fish practice	29	12.0
	Do not practice	15	6.2
	Other practice	11	4.6
	Exotics practice (parrots, tortoise, snakes, etc.)	8	3.3
	Wildlife practice (wild animals)	6	2.5
Location of veterinary practice by geopolitical zone	North Central	69	28.6
	North West	49	20.3
	North East	46	19.1
	South West	39	16.2
	South East	21	8.7
	South South	17	7.1

* The results for this variable are presented as row percentages instead of column percentages, as such the column percentages for this variable do not add up to 100%.

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
