# Peer review of "Knowledge, Attitudes and Practices of Veterinarians Towards Antimicrobial Resistance and Stewardship in Nigeria"

_antibiotics, 2020, doi:10.3390/antibiotics9080453_

Round 1

Reviewer 1 Report

Interesting study that highlights an important topic

The authors describe the results of a cross sectional survey on knowledge, attitudes and practices of veterinarians in Nigeria

The methodology and results are described thoroughly

Suggestions to improve the manuscipt

  1. The demographics could be presented via table 1 only. The text currently offers information that is presented i  the table and the manuscipt is too lenghty
  2. Please add a paragraph on limitations of the present study. For example a very low response rate in the survey across the country. Are there any possible explanations for that?
  3. Apart from veterinary education the authors could add more possible solutions to the problem such as for instance governmental interventions  to cancel bying antibiotics without prescription

Author Response

Reviewer 1

Comments and Suggestions for Authors

Interesting study that highlights an important topic

The authors describe the results of a cross sectional survey on knowledge, attitudes and practices of veterinarians in Nigeria

The methodology and results are described thoroughly

The authors thank the reviewer for taking time to read the manuscript and provide comments. The suggested changes and comments were constructive and have helped improve the quality of the manuscript.

Revisions have been made to the manuscript using track changes. Please find below the point-by-point response to each comment and corresponding line numbers. Reviewer comments are in bold and text with author responses is in italics.

Suggestions to improve the manuscipt

  1. The demographics could be presented via table 1 only. The text currently offers information that is presented i  the table and the manuscipt is too lengthy

The authors agree that text describing demographics could be further reduced since similar data are provided in detail in Table 1. The authors have further reduced the amount of text in this section considering details are provided in Table 1. Please see summary of Table 1 in Lines 119-125.

“Most of the respondents were male (79.7%, 192/241) and almost half were aged 25-34 years (48.1%, 116/241) (Table 1). Majority of respondents reported having a veterinary degree (63.1%, 152/241) as the highest educational qualification. More than a third of respondents had been registered as a veterinarian for 0-5 years (36.1%, 87/241). Mixed practice (defined as any combination of small, large, poultry or other type of practice) was most frequently reported type of practice (63.5%, 153/241). Five percent of respondents, (5.4%, 13/241) did not practice. Most of the respondents were employed in private practice (38.2%, 92/241).”

  1. Please add a paragraph on limitations of the present study. For example a very low response rate in the survey across the country. Are there any possible explanations for that?

The limitation of low response rate had been discussed in the manuscript under the discussion paragraph with limitations and has been revised further. Please see Lines 420-431.

“A few limitations were observed during the conduct of the current study. The response rate of the survey was low, and as such, study findings may not be generalizable to the whole of the country; however, the gaps identified can still inform the development of interventions targeting all veterinarians in Nigeria. The low response rate may have been due to several factors such as unwillingness to participate or lack of internet access in some parts of the country. There may also have been selection bias in the respondent population. For example, it is possible that respondents that completed the survey were more technologically astute or inclined. It could also be that these respondents had a special interest in the subject, hence their participation. Another potential limitation was social desirability bias which might have affected the nature of responses provided; it’s possible that some respondents may have declined to share information they considered inappropriate or erroneous, resulting in an under-reporting of certain aspects on antibiotics and AMR knowledge and practices.”

  1. Apart from veterinary education the authors could add more possible solutions to the problem such as for instance governmental interventions to cancel bying antibiotics without prescription

Authors agree that government interventions such as through development and implementation of relevant policies and regulations and enforcement of regulations is one of the potential solutions worth exploring. This has been further discussed and added to the conclusion section. Please see Lines 407-410.

“In addition, government interventions such as formulation and implementation of relevant policies and regulations and enforcement of such regulations may also be useful in improving appropriate antibiotic use and stewardship.”

Reviewer 2 Report

A very insightful and important study. Please note line 114 Results contains journal instructions, same for line 568 acknowledgements. 

Author Response

Reviewer 2

The authors thank the reviewer for taking time to read the manuscript and provide comments. The suggested changes and comments were constructive and have helped improve the quality of the manuscript.

Revisions have been made to the manuscript using track changes. Please find below the point-by-point response to each comment and corresponding line numbers. Reviewer comments are in bold and text with author responses is in italics.

Comments and Suggestions for Authors

A very insightful and important study. Please note line 114 Results contains journal instructions, same for line 568 acknowledgements.

Corrections made, instructions deleted. Please see Line 113 (Results section) and Line 553-561 (acknowledgements).

Reviewer 3 Report

Adekanye et al present results of a survey of the knowledge, attitudes and practices of veterinarians toward Antimicrobial Resistance and Stewardship in Nigeria. In my opinion the approach is sound, the manuscript is well written, and the information is valuable to the scientific community. It is unfortunate that the survey response rate was so low, so while that affects inferences that can be made from these data, it does not preclude publication. I have only one minor comment for consideration, at line 114 to 116 I believe the instructions to authors were inadvertently included in the text.

Author Response

Reviewer 3

The authors thank the reviewer for taking time to read the manuscript and provide comments. The suggested changes and comments were constructive and have helped improve the quality of the manuscript.

Revisions have been made to the manuscript using track changes. Please find below the point-by-point response to each comment and corresponding line numbers. Reviewer comments are in bold and text with author responses is in italics.

Comments and Suggestions for Authors

Adekanye et al present results of a survey of the knowledge, attitudes and practices of veterinarians toward Antimicrobial Resistance and Stewardship in Nigeria. In my opinion the approach is sound, the manuscript is well written, and the information is valuable to the scientific community. It is unfortunate that the survey response rate was so low, so while that affects inferences that can be made from these data, it does not preclude publication. I have only one minor comment for consideration, at line 114 to 116 I believe the instructions to authors were inadvertently included in the text.

Correction made. Please see Line 113.

Reviewer 4 Report

Dear Authors,

the topic you have dealt with is very interesting not only for the experts in the sector but also for further raising awareness among the entire Veterinary Public Health sector regarding the problem of Antibiotic Resistance. I also think that currently it is not yet ready for publication but only a few small changes are needed.

Below are my comments and suggestions for improving your manuscript.

Comments and Suggestions for Authors

Line 38 (Keywords): the keywords are repetitive and should never be the same as indicated in the title. I suggest modifying them above all to increase the availability of the article.

Line 113 (Results): Although this section is very well written, the tables are too many and the data entered unnecessarily overlap with those reported in writing. Please modify. In addition, I suggest to summarising the tables or eventually moving them in the supplementary materials.

Line 282: The style of the “Discussion” should be bold and without “.0”. Please edit.

In addition, I suggest to the Authors to structure a single discussion and to create a subsection only for the limits of the study.

Lines 531-556: The conclusion is written in a confusing way and mixed with the results of the study. I think it should be modified.

References

The references must be partially changed, since they were not written following the “instructions for authors" of the journal.

Author Response

Reviewer 4

Comments and Suggestions for Authors

Dear Authors,

the topic you have dealt with is very interesting not only for the experts in the sector but also for further raising awareness among the entire Veterinary Public Health sector regarding the problem of Antibiotic Resistance. I also think that currently it is not yet ready for publication but only a few small changes are needed.

Below are my comments and suggestions for improving your manuscript.

The authors thank the reviewer for taking time to read the manuscript and provide comments. The suggested changes and comments were constructive and have helped improve the quality of the manuscript.

Revisions have been made to the manuscript using track changes. Please find below the point-by-point response to each comment and corresponding line numbers. Reviewer comments are in bold and text with author responses is in italics.

Comments and Suggestions for Authors

Line 38 (Keywords): the keywords are repetitive and should never be the same as indicated in the title. I suggest modifying them above all to increase the availability of the article.

Key words revised. Please see Line 38.

Line 113 (Results): Although this section is very well written, the tables are too many and the data entered unnecessarily overlap with those reported in writing. Please modify. In addition, I suggest to summarising the tables or eventually moving them in the supplementary materials.

To reduce overlap, Tables 2-4 have been moved to Appendix. Please see Tables A3-A5.

Line 282: The style of the “Discussion” should be bold and without “.0”. Please edit.

In addition, I suggest to the Authors to structure a single discussion and to create a subsection only for the limits of the study.

Correction made. Please see Line 265.

Subheadings in the discussion section were removed.

Lines 531-556: The conclusion is written in a confusing way and mixed with the results of the study. I think it should be modified.

Following the journal formatting requirements, the conclusion was placed at the end of the results section in the original version. Authors agree, for clarity, the conclusion is better placed just after the discussion section, but for purposes of satisfying journal formatting requirements, the authors have left the conclusion in the same location. Minor editorial edits were made to the conclusion to further clarify the text, see Lines 516-541.

References

The references must be partially changed, since they were not written following the “instructions for authors" of the journal.

Edits made to references.